# Platinized graphene fiber electrodes uncover direct spleen-vagus communication

Maria A. Gonzalez-Gonzalez[1], Geetanjali S. Bendale[1], Kezhong Wang[2], Gordon G. Wallace[2] & Mario Romero-Ortega [1✉]

Neural interfacing nerve fascicles along the splenic neurovascular plexus (SNVP) is needed to better understand the spleen physiology, and for selective neuromodulation of this major organ. However, their small size and anatomical location have proven to be a significant challenge. Here, we use a reduced liquid crystalline graphene oxide (rGO) fiber coated with platinum (Pt) as a super-flexible suture-like electrode to interface multiple SNVP. The Pt-rGO fibers work as a handover knot electrodes over the small SNVP, allowing sensitive recording from four splenic nerve terminal branches (SN 1–4), to uncover differential activity and axon composition among them. Here, the asymmetric defasciculation of the SN branches is revealed by electron microscopy, and the functional compartmentalization in spleen inner-vation is evidenced in response to hypoxia and pharmacological modulation of mean arterial pressure. We demonstrate that electrical stimulation of cervical and sub-diaphragmatic vagus nerve (VN), evokes activity in a subset of SN terminal branches, providing evidence for a direct VN control over the spleen. This notion is supported by adenoviral tract-tracing of SN branches, revealing an unconventional direct brain-spleen projection. High-performance Pt-rGO fiber electrodes, may be used for the fine neural modulation of other small neurovascular plexus at the point of entry of major organs as a bioelectronic medical alternative.

[1] Biomedical Engineering and Biomedical Sciences, University of Houston, Health 2, 4849 Calhoun Rd., Room 6014, Houston, TX 77204-6064, USA. [2] Intelligent Polymer Research Institute, ARC Centre of Excellence for Electromaterials Science, University of Wollongong, Wollongong, NSW 2522, Australia. ✉email: miromeroortega@uh.edu

Vagus nerve stimulation (VNS) has been shown to reduce the circulation of inflammatory cytokines and to improve survival during severe inflammation in rodents[1–3]. The immunosuppression effect of VNS has been proposed to be mediated by activation of the anti-inflammatory reflex pathway, where brain stem vagal complex nuclei forming the vagus nerve (VN) innervate the celiac ganglia (CG), which in turn, forms the splenic nerve (SN), which containing predominantly sympathetic vasoconstrictor fibers[4,5], and mediating the release of acetylcholine (Ach) from T-cells in the spleen[6,7]. These studies led to the notion that VNS can be used as a treatment for severe immune disorders, including inflammatory bowel disease and rheumatoid arthritis[2,8]. However, the idea of a direct cholinergic innervation to the spleen is a subject of much debate and was directly challenged by Bratton and collaborators, who failed to confirm a direct SN synaptic activation by VNS[9,10]. The controversy is also based on the apparent lack of direct parasympathetic input to the spleen and the consideration that its neural control is exclusively sympathetic[11]. It is known that efferent norepinephrine (NE) fibers travel along arterial and apical branches to innervate lymphocytes and macrophages in the spleen[12,13]. However, a small percentage of Ach fibers are located apically[14] and seem to represent afferent axons from vein baroreceptors projecting to the nucleus of the tractus solitarious (NTS) in the brain stem, which responds to changes in blood pressure (BP)[15–17]. Indeed, the SN contains axons from different peptidergic neurons that were proposed to differentially modulate spleen vascular functions[18,19]. Part of the challenge in the study of this organ is the small size of the rat SN (i.e., ~100 μm diameter) composed of 300–400 axons in 3–6 fascicles traveling along blood vessels, and forming a neurovascular plexus (SNVP), that branches before entering the 2–3 lobular organ[4,20]. This makes the visualization and isolation of the SN for electrophysiology or tract-tracing studies, highly challenging.

We reasoned that the functional understanding of the complex innervation to the spleen will require simultaneous and independent recording of the four splenic terminals neurovascular plexus (SNVP-1 to 4) during evoked physiological events. However, current silver or platinum hook, or cuff electrodes, lack the miniaturization, flexibility, sensitivity, or charge injection capacity to allow effective communication with these small nerve fascicles[14,21]. We recently developed a reduced liquid crystalline graphene oxide (rGO) fiber coated with a thin platinum (Pt) layer to produce a highly sensitive and super-flexible Pt-rGO electrode with unmatched mechanical and electrochemical characteristics[22]. The high geometric surface area of these fiber electrodes provides unrivaled charge injection capacity (10.34 mC/cm$^2$) and low impedance (28.4 ± 4.1 MΩ μm$^2$; 1 kHz), superior to conventional carbon nanotube arrays or other common materials used as neural interfaces[23,24], and able to record cortical neural activity with a signal to noise ratio (SNR) of 9.2 dB. Here, we report that Pt-rGO fibers can be used as an overhand knot suture electrode (aka "sutrode") around individual SNVP, without isolating the SN terminal branches from the adjacent vasculature, to effectively and sensitively interface this nerve. Transmission electron microscopy (TEM) of the SN branches revealed asymmetric defasciculation from the parent nerve, and simultaneous neural recording from all terminal SN branches using the sutrode, uncovered functional compartmentalization in the spleen innervation in response to hypoxia and pharmacological modulation of BP. Furthermore, electrical stimulation of cervical and sub-diaphragmatic VN (cVN and SD-VN), evoked direct activity in a subset of SN terminal branches, providing evidence for a direct VN control over the spleen. This notion was supported by tract-tracing of apical and basal SN branches, revealing an unconventional direct brain-spleen projection. The

use of high-performance Pt-rGO fiber electrodes, provided functional information of individual small SNVP, shedding light on the fine neural control of organ physiology and enabling the neuromodulation at the organ point of entry.

## Results

**Use of Pt-rGO fibers as "sutrodes".** The flexibility and mechanical robust nature of the Pt-rGO fibers allowed their use as suture electrodes. To that end, we made a slight modification to the previously reported fabrication method[22], masking 1 cm proximal and distal segments of 10 cm Pt-rGO fibers before Parylene C coating. The proximal end was soldered to a Pt wire using silver (Ag) paste for connection to amplifiers, and the distal end was used for recording/stimulation (Fig. 1a–a′). Cyclic voltammetry was used to confirm that the electrochemical characteristics of the Pt-rGO fibers were not affected by these changes (Fig. 1b). We then measured the impedance of the sutrode while tying a knot to complete closure, and confirmed the flexibility and conductivity of the fiber electrode (Fig. 1c; Supplementary Video 1). To demonstrate the mechanical robustness of the sutrode, we tied it to a 9–0 polyamide suture needle, and successfully drove the sutrode through the rat biceps femoris muscle without breaking (Fig. 1d, f). As expected, applying a 2–10 V potential, evoked visible graded muscle contractions (Supplementary video 2). The noise measured by the sutrode was lower (33.3 ± 2.3 μV) compared to that sensed by a conventional Pt hook electrode (56.7 ± 6.6 μV, $p < 0.01$) (Fig. 1e). We then evaluated the use of the sutrode as a recording electrode by wrapping it around the sciatic nerve (ScN), tying a knot over it carefully, not to occlude the epineural blood circulation (Fig. 1g), and separately, used as a monopolar electrode to record the evoked compound action potential from muscle sensory afferents as the muscle contracted (Fig. 1h). We then placed the sutrode around the tibial nerve fascicle and using a stimulating hook electrode proximally, demonstrated the recording of compound nerve action potentials (CNAPs) evoked at different voltage strengths (Fig. 1i, j). In addition, and consistent with our previous report in the cerebral cortex, we also confirmed that multiple individual sutrodes can be inserted directly into peripheral nerves for a sensitive recording of intraneural single units with an SNR = 9.6 (Supplementary Fig. 1). These tests confirmed the use of the sutrode as a sensitive overhand knot electrode for modulation of the peripheral nerve.

**Sutrode recording of physiologically evoked activity in the cVN.** We tested the ability of the sutrode to record spontaneous and evoked physiological activity in the cVN using the Pt-rGO fiber as a hook electrode. Stable baseline activity was recorded with 20 μV peak-to-peak (pp; biological noise) and spontaneous waveforms of 30–60 μV pp at 1–3 spikes per second were identified. Raster plots, principal component analysis (PCA), and interspike interval analysis are presented in Supplementary Fig. 2. We then induced hypotension by the intravenous administration of the nitric oxide donor sodium nitroprusside (NPS) to evaluate the evoked neurophysiological responses in the cVN (Fig. 2a). After ~1 min of NPS administration, the mean arterial blood pressure (MAP) decreased by 50 mmHg approximately. This coincided temporally with a reduction in the amplitude of spontaneous activity in vagal activity lasting 20 s, which was followed by the appearance of CNAPs activity of 80–100 μV waveforms at high frequency (Fig. 2b), possibly from baroreceptor and cardiovascular afferents. To confirm the ability to record physiological activity from different types of axons in the cVN, changes in spontaneous activity were evoked by 2 min lapse in oxygen restriction. Figure 2c shows a representative recording of basal spontaneous activity of an isolated compound action potential waveform (Fig. 2d). This activity decreased immediately

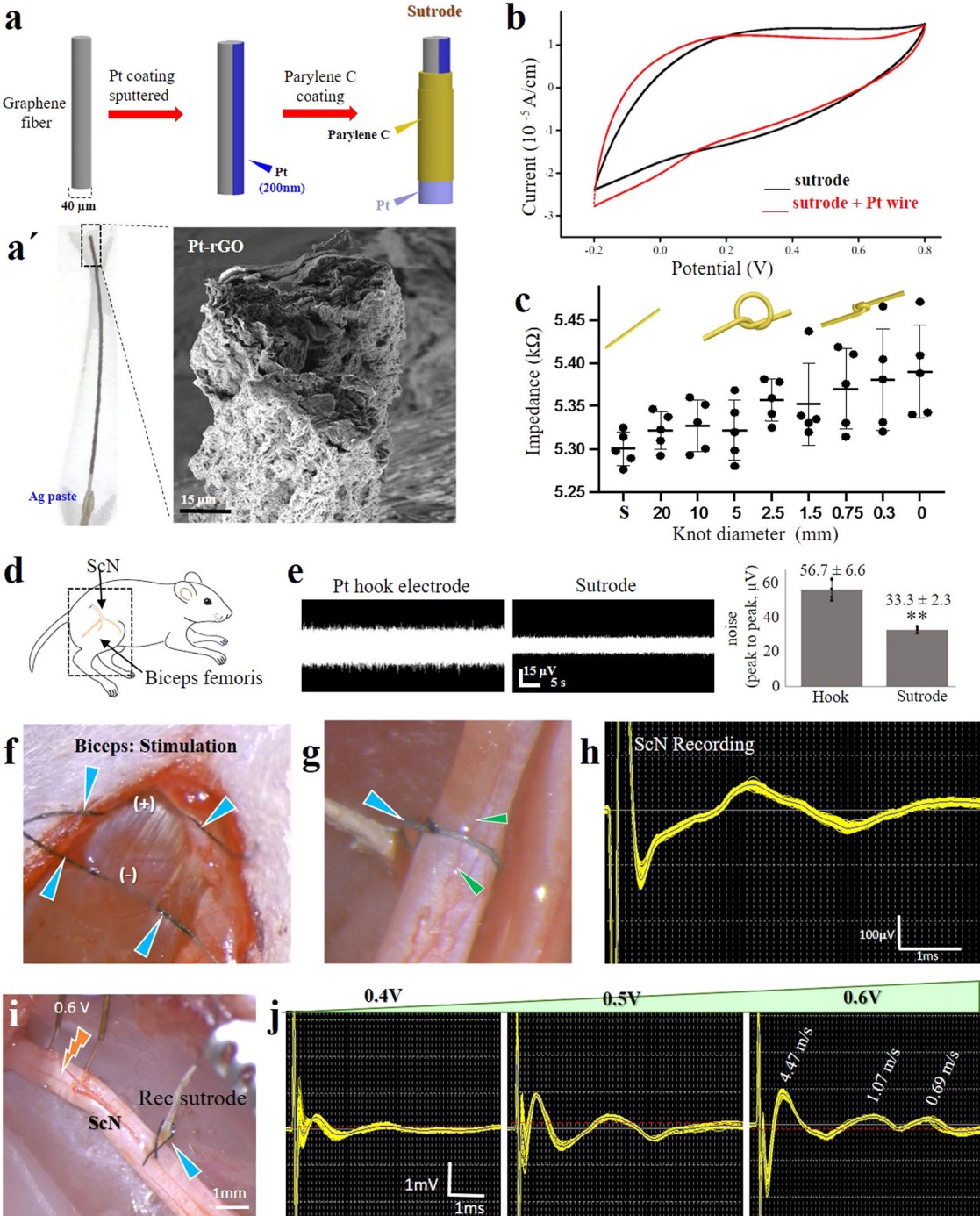

**Fig. 1 Platinized graphene fibers serve as suture and wrapping electrodes. a** Coating steps of extruded Pt-rGO electrodes. **a′** Photograph and SEM. **b** Cyclic voltammetry of Pt-rGO fibers before and after soldering to a Pt wire. **c** Graphene fiber knotting does not significantly alter the electrode impedance. Individual impedance measurements and their mean and standard deviation are represented in the graphic during different knot diameters in the sutrode ($n = 5$ sutrodes). **d** Illustration of the ScN and biceps femoris muscle. **e** Reduced base noise measurements of the sutrode compared to Pt hook electrodes in saline, graphics in the right were obtained from $n = 5$ independent experiments **$p < 0.01$ (values represented as mean ± SD). **f** Sutrode placed on biceps (blue arrows) evoked effective muscle contraction (Supplementary Video 1). **g** Sutrode wrapped snugly on the ScN (blue arrow), without blood flow occlusion of superficial vasculature (green arrows) used to **h** record the evoked CNAP. **i** Stimulation of the tibial nerve fascicle with hook electrodes and recording with the sutrode (blue arrow), allowed the recording of **j** graded evoked CNAPs and the detection of B and C fibers. Pt-rGO reduced liquid crystalline graphene oxide, s straight sutrode, ScN sciatic nerve, SEM scanning electron microscopy, CNAP compound nerve action potentials.

after the hypoxia, but 180 s later, a new CNAP was detected firing at 10-fold higher frequency compared to baseline, likely the result of cardio-pulmonary reflex activity, as it was reversed by normoxia (Fig. 2c, d). The plotting of multiple independent measurements revealed a maximum peak of increase in activity from

$6.5 \pm 3.1$ to $15.3 \pm 2.7$ spikes $10\,s^{-1}$ (graphic in Fig. 2d, $p < 0.01$). The initial decrease in cVN activity one minute after oxygen restriction correlated with a mild reduction in heart rate (from 348 to $346 \pm 4.1$ beats per minute) followed by a mild increase in breathing rate (from 42 to $46 \pm 2.8$ breaths per minute) (Fig. 2e,

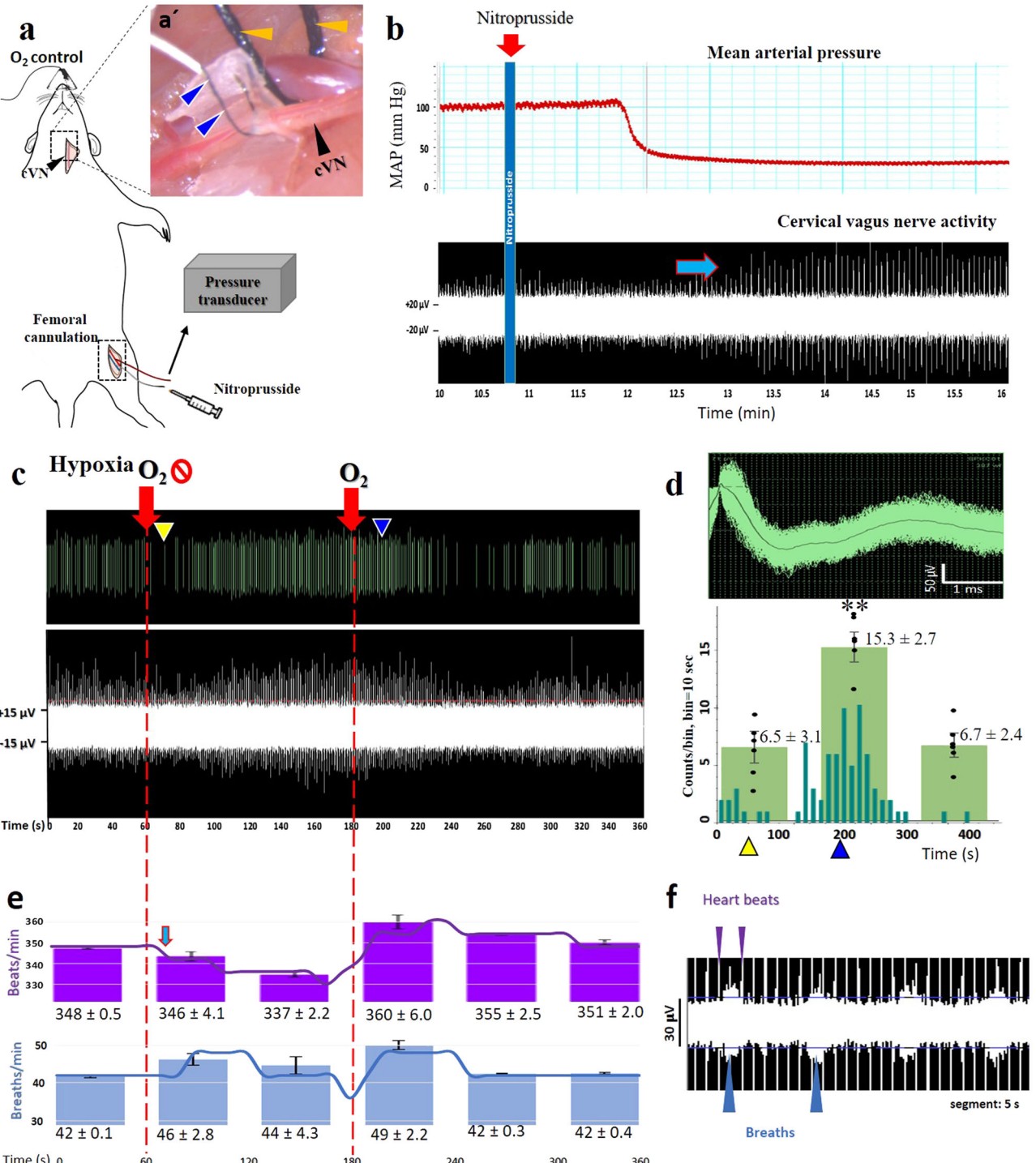

**Fig. 2 Vagal neuronal activity evoked by hypotension and hypoxia recorded by sutrode. a** Schematic of sutrode placement on the rat cVN, with a femoral arterial blood pressure sensor. **a′** The cVN (black arrow) was interfaced with a sutrode (blue arrows). A conventional 4-0 nylon suture was used to isolate the carotid artery (yellow arrows). **b** Nitroprusside (red arrow) induced hypotension, that correlated with an increase in cVN firing rate (blue arrow). **c** Oxygen restriction induced an initial decrease (yellow arrowheads) and a subsequent 10-fold increase (blue arrowheads) in vagal activity. **d** Representative rate histogram of an isolated waveform (dark green bars), and graph (overlapped) of average total neuronal activity $n = 6$ experiments from $n = 3$ rats (averages ± SD), whiskers represent confidence intervals **$p < 0.01$. Oxygen restriction and restoration are pointed as red arrows in **c**. **e** The Oxygen reduction corresponded to a slight decrease in heart rate (blue arrow), and a subsequent increase in respiration rate calculated from **f** unfiltered raw data files. Values are from a representative rat, and overlapped bars are presented as the average ± SD of three rats.

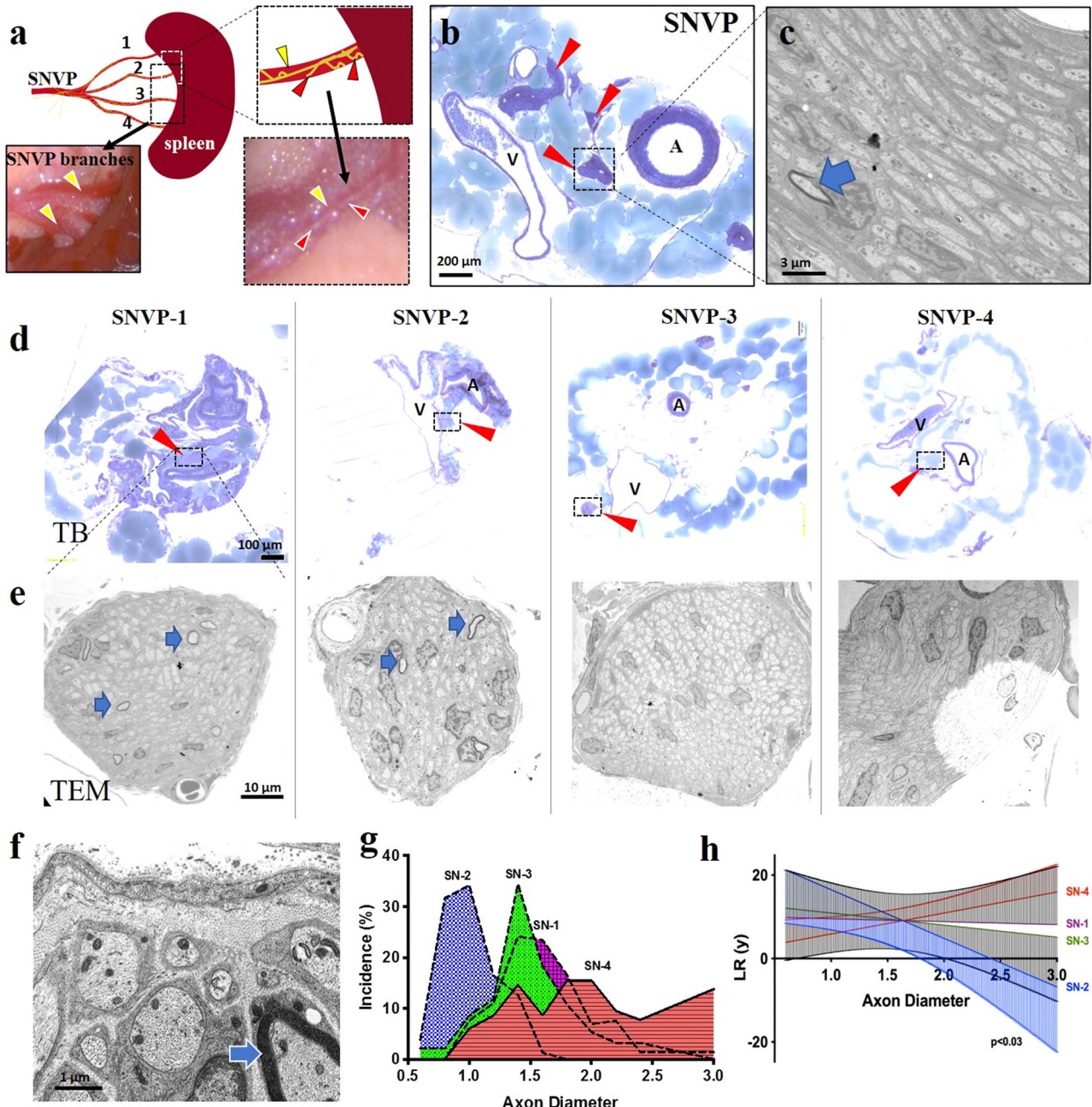

**Fig. 3 Splenic neurovascular plexus asymmetric branching. a** Schematic of the rat SNVP splitting into four-terminal branches prior to entering into the spleen. Inserts: Splenic vasculature (yellow arrowheads) and the SN (red arrowheads). **b** Toluidine blue (TB) staining, red arrows point to the location of the SN fascicles between the artery (A) and vein (V). **c** unmyelinated SN axons by TEM with a single myelinated axon (blue arrow). **d** Representative images of TB staining of the four-terminal SN branches, with nerve fascicles in inserts (red arrowheads). **e** TEM micrographs of axonal composition in each terminal branch. Myelinated axons (blue arrows) were observed in SN1–2, but not in SN 3–4. **f** High TEM magnification of SN-2 axons. **g** Axon diameter distribution in the SN branches. **h** Linear regression analysis of axon diameter distribution. Kolmogorov–Smirnov normality test was used to determine the difference in axon diameter population in different groups ($n = 4$ animals). Data are shown as mean (solid lines) and error (95 confidence interval; top for SN1 and SN4, and bottom for SN3 and SN2). SN1–4 are shown by different colors. SN splenic nerve, SNVP splenic neurovascular plexus, TEM transmission electron microscopy. The slope between SN-1 and SN-4 is significantly different ($p < 0.03$).

f), suggesting the initial reduction in baroreceptor signals and subsequent recruitment of lung Aδ/B mechanoceptors and nociceptors. Together, the data confirmed the use of the sutrode as a sensitive electrode for extraneural recording of physiological compound action potentials in the cVN.

**Asymmetric axon branching into terminal splenic neurovascular plexus.** Functional studies on the neural control of the

spleen are limited by the location of the SN, as it travels along the vasculature and splits into terminal branches before entering the organ (Fig. 3a), and the incomplete anatomical description of the axonal fasciculation. To learn about the axon composition in the SN branches, we did histological and electron microscopic evaluations of the terminal four SNVP. We observed multiple fascicles in the SN ranging from 100 to 500 μm in diameter, located between the splenic artery and vein, and composed of a

heterogeneous population of axon types with the majority being unmyelinated (Fig. 3b, c). The SN fascicles split along the vascular branches into 1–2 fascicles (50–100 μm in diameter) that followed the splenic vasculature, surrounded by fatty tissue (Fig. 3d, e). Morphometric analysis of the SNVP revealed that the splitting of SN axons into the terminal branches is notably asymmetric (Fig. 3f–h), where the average diameters of unmyelinated axons were $1.70 \pm 0.4$ μm in SN-1, $0.92 \pm 0.2$ μm in SN-2, $1.38 \pm 0.4$ μm in SN-3, and $1.89 \pm 0.4$ μm in SN-4 ($n = 4$). Furthermore, few (3–6) myelinated axons were present only in SN-1 (median 2.54 μm; 0.85 g-ratio) and SN-2 (median 0.95 μm; 0.71 g-ratio), but not in SN-3,4. The different axon composition is evident in Fig. 3g, where the D'Agostino–Pearson normality test revealed that axon diameters in SN-1, SN-2, and SN-4 showed normal distribution in axon diameter ($p = 0.45$, 0.15, and 0.66; respectively), whereas SN-3 did not ($p < 0.003$). Linear regression and goodness to fit test were done to evaluate if the axon population of the SN branches were dissimilar. Figure 3h shows the linear regression of SN1–4 (solid lines) and the error (95 confidence interval; top for SN1 and SN4, and bottom for SN3 and SN2). The slope of SN-1 ($-8.83$ to 8.4; $R^2 = 0.003$) and SN-2 ($-21.88$ to $-1.45$; $R^2 = 6.67$) showed significant differences, and only SN-2 significantly deviated from zero ($F = 6.68$, DFn $= 1.9$, $p < 0.03$). Together, the data indicated that the axon composition of the terminal branches is not homogeneous.

**Sutrode interfacing of splenic terminal neurovascular plexus (SNVP) revealed differential physiology**. The heterogeneous axonal composition in the SNVP compelled us to use the sutrode to record simultaneously from the four-terminal branches and investigate if their neural activity varied in response to specific physiological stimuli. To that end, we made a handover knot with the sutrode to wrap and interface each terminal splenic branch (Fig. 4a, a′). Despite the small size (50–100 μm) and inter-vessel location of these nerve fascicles, we were able to record spontaneous activity with great sensitivity (SNR $= 8.5 \pm 0.2$ pp, Fig. 4b, b′). We used PCA on the unfiltered data to identify neural waveforms of spontaneous activity in a 6.5 ms window. Raster plot of the selected units (Fig. 4c, d) showed distinct waveforms ranging from 50 μV pp in SN-1,3 to 400 μV pp in SN-4 (Fig. 4d). The pattern of spontaneous activity in the four SN terminal branches was different, with tonic activity observed in SN-1 and SN-3, but not in the other two branches.

In response to hypoxia, the activity in SN-1 increased in frequency, whereas SN-2 showed a sharp decrease in activity immediately after hypoxia (Fig. 4c and Supplementary Fig. 3). In order to confirm the differential functional response of the SN terminal nerve fascicles to physiological events, we used the sutrode to record the response to NPS-induced vasodilation and hypotension, from the cVN and the SN terminal branches, simultaneously (Fig. 5). Consistent with the differential axonal composition, we were able to discriminate multiple CNAP waveforms in the cVN and the SN terminal branches (Supplementary Fig. 4), with unique temporal resolution. Immediately after NPS administration, a tonic active waveform in the cVN (Wf-a, Fig. 5a) drastically reduced its firing activity (Fig. 5b), this event was followed by a delayed increase in Wf-b, and a late appearance of Wf-c 250 s later. Considering a threshold of $0.4$ spikes s$^{-1}$ to interpret significant changes in activity frequency, the SN-1 nerve responded immediately after NPS and continued to increase in frequency over time. The activity levels of SN-2 and SN-4 showed an increase in frequency that coincided with that of Wf-b in the cVN (yellow dotted line Fig. 5b–d). In contrast, neural activity in SN-3 did not change substantially. The differential in baseline waveform peak-to-peak amplitude in a 200 s segment, was compared to that evoked by

NPS vasodilation. The evoked activity in SN-1 to NS-4 resulted in delta increment of $4.6 \pm 1.3$, $3.4 \pm 1.9$, $0.9 \pm 0.8$ and $2.8 \pm 1.2$ μV, for each terminal branch; respectively. Tukey's multiple comparisons test evidenced a significant difference between SN-1 and SN-3 (**$P = 0.006$, alpha $= 0.05$, $n = 5$ animals, see Supplementary Fig. 4). Together, the physiological response from the individual SN branches suggests that they carry different functional information to and from the spleen, in a unique temporal pattern.

**VNS evokes differential activity in SN terminal branches**. The apparent temporal correlation between induced activity in the VN and the terminal SN branches indicated a direct VN control on the spleen. Such a functional connection between the VN and the spleen has been suggested before, but it is currently a topic of controversy. Given the differential anatomical and functional activity in the four SN terminal branches, we reasoned that the functional connection of the VN to the spleen might be related to some, but not all axons in the SN, possibly explaining the controversial reports. To address this question, we recorded the neural activity from the four SNVP in response to cVN stimulation. The application of 30 s stimulation of 0.5 V pulses, evoked an immediate 3-fold increase in nerve frequency and amplitude in SN-1 and approximately a 2-fold increase in SN-2 branches, but no changes were seen in SN-3 and SN-4 (Fig. 6a, b). These variations in recorded activity were different between branches ($p < 0.01$). This activity was blocked with lidocaine applied to the splenic branches confirming the neural nature (Fig. 6c). The differential response was also evidenced by increasing the stimulation voltage. At 1.2 V, the activity in the VN was induced and was followed by the increase of neural activity in SN-1 and SN-2. Activity in SN-3 only increased above 1.6 V stimulation and, in contrast, that of SN-4 decreased (Supplementary Fig. 5).

In order to better establish the functional relationship of the VN and the SN terminal branches, we used sutrodes to record from the SN-1 in response to stimulation of the cVN and SD-VN (Fig. 7). We confirmed that activation of the cVNS evokes the increase in frequency firing of the SN-1 branch (Fig. 7a). However, increasing the depolarization voltage to 1.5 V did not seem to have the same effect, even if the pulse duration is increased to 0.5 ms. Importantly, SD-VN stimulation at 80 mV for 0.2 ms pulses also induced an increase of activity in SN-1. This activation was parameter specific, since a progressive increase in stimulation voltage to 100 mV resulted in inhibition of the evoked response (Fig. 7b). Analysis of VNS evoked amplitude changes in SN-1 compared to baseline activity (peak-to-peak; $n = 3$; 3–5 replicates per animal), showed that 0.5 V, 0.3 ms pulses evoked an increase in $18.16 \pm 1.6$ μV, while lowering the stimulation to 100 mV, 0.2 ms pulses, resulted in $-10.68 \pm 3.07$ μV amplitude decrease. Student $t$-test between these groups showed a significant difference between these two groups $P < 0.0001$ (Fig. 7c). These results underscore the importance of parameter specification in the control of activity of the splenic terminal plexus. Despite that SD-VNS evoked changes in terminal branch activity that developed over seconds, the first evoked waveform appeared 1–3 ms in SN-1 (Supplementary Fig. 6), suggesting the possibility that this branch directly connects the VN to the spleen.

**Tracing of VN fibers to the splenic terminal plexus**. To confirm the possibility that SN-1 contains axons from the cVN, we used viral tract-tracing and microinjected an adenovirus (AdV) tracer encoding the green fluorescent protein (Ad-GFP) into the SN-1, and a second AdV tracer with the red fluorescent label mCherry into SN-3 (Ad-mCherry) (viral construction is presented in Supplementary Fig. 7), and looked for labeled axons in the cVN 6 days

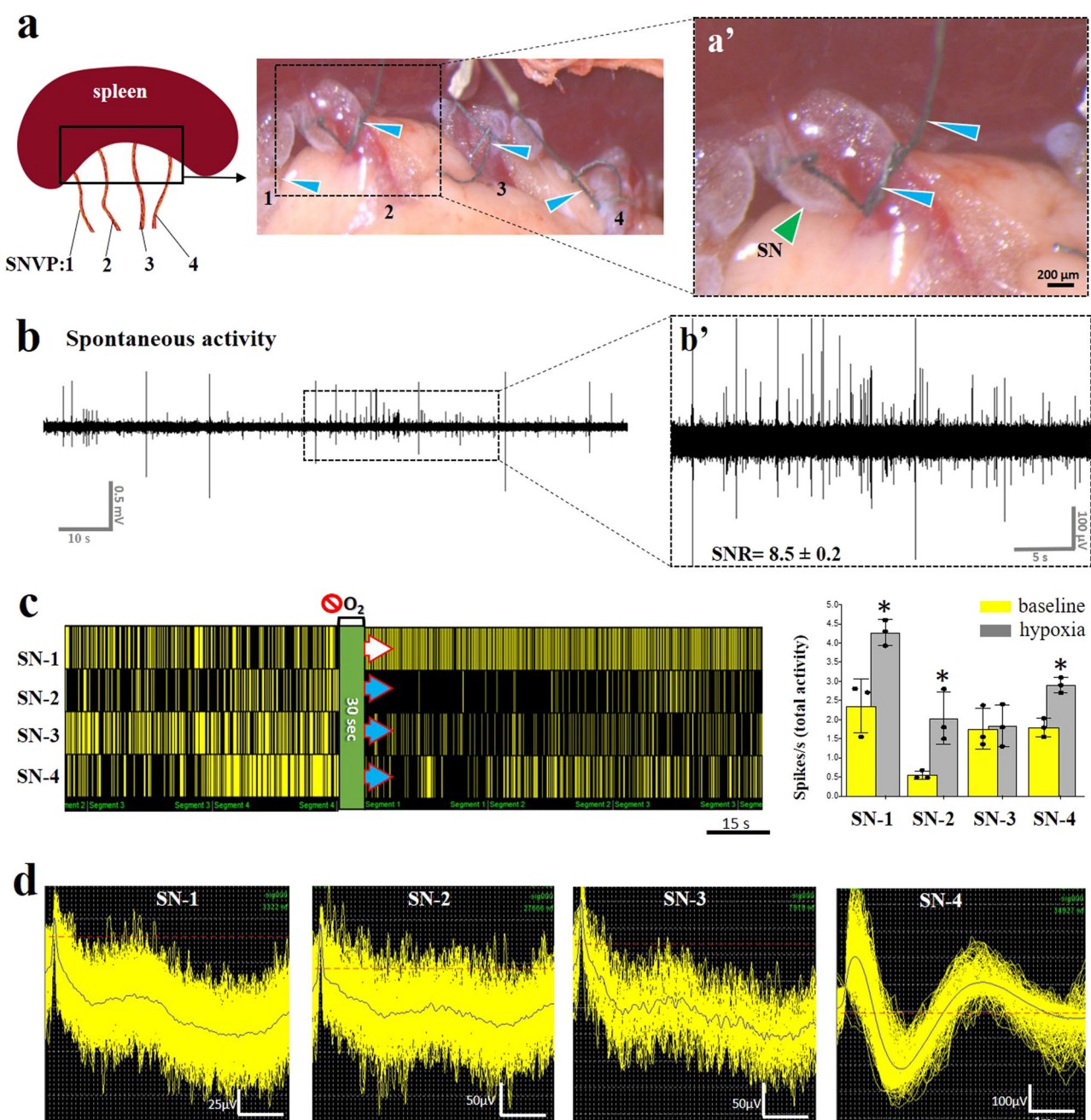

**Fig. 4 Sutrode recording of SNVP terminal branches revealed differential response to Oxygen deprivation. a** Schematic and pictures of the SNVP branches (1–4; apical to basal) interfaced with sutrodes (blue arrowheads in **a** and **a´**), tied around the SNVP (**a´**), green arrowhead points out a SN branch. **b** Representative raw spontaneous recording of neural activity from SN-1 (signal to noise ratio, SNR = 8.5 ± 0.2 peak to peak, **b´**. **c** Simultaneous raster plots from a representative waveform from the four SN branches showing differential patterns of activity before and after Oxygen deprivation (green strip) of identified CNAPs waveforms, on the right, overlapped graphics of total activity, comparing baseline (yellow), and at 1 min after Oxygen deprivation (gray) for all the SN, $n = 3$), whiskers represent SD, *$p < 0.05$. **d** The spontaneous activity increased in SN-1 (white arrow in **c**) but decreased in SN-2-4 (blue arrows in **c**). SN splenic nerve, SNVP splenic neurovascular plexus, SNR signal to noise ratio, CNAPs compound nerve action potentials.

after (Fig. 8a). The expression of the GFP biomarker was confirmed in the spleen, where clusters of small axons were observed (Fig. 8b, c´). This method confirmed the labeling of sparsely distributed and clustered GFP axons in the cVN (Fig. 8d and Supplementary Fig. 9d, e and 10). We consistently observed traced axons from the terminal spleen branches in all 7 animals and estimated a population of 128 ± 39 axons per branch. Wilcoxon test shows no significant difference in the distribution of the number of labeled SN-1 axons from the mean, ð›¼ = 0.05,

Fig. 8e), confirming the reproducible labeling of these neurons in the VN. Evaluation of the brain stem in these animals, revealed GFP-positive neurons in the dorsal motor nucleus (DMN) of the vagal complex, indicating that some of the axons in the SN-1 are efferent extensions of parasympathetic central neurons (Fig. 8f, g´´ and Supplementary Fig. 9g–i). A similar pattern was observed with the Ad-mCherry label, the expression of the fluorescent protein was confirmed in the spleen and small axons were seen in the cVN fascicles, albeit in very small numbers and not easily detected

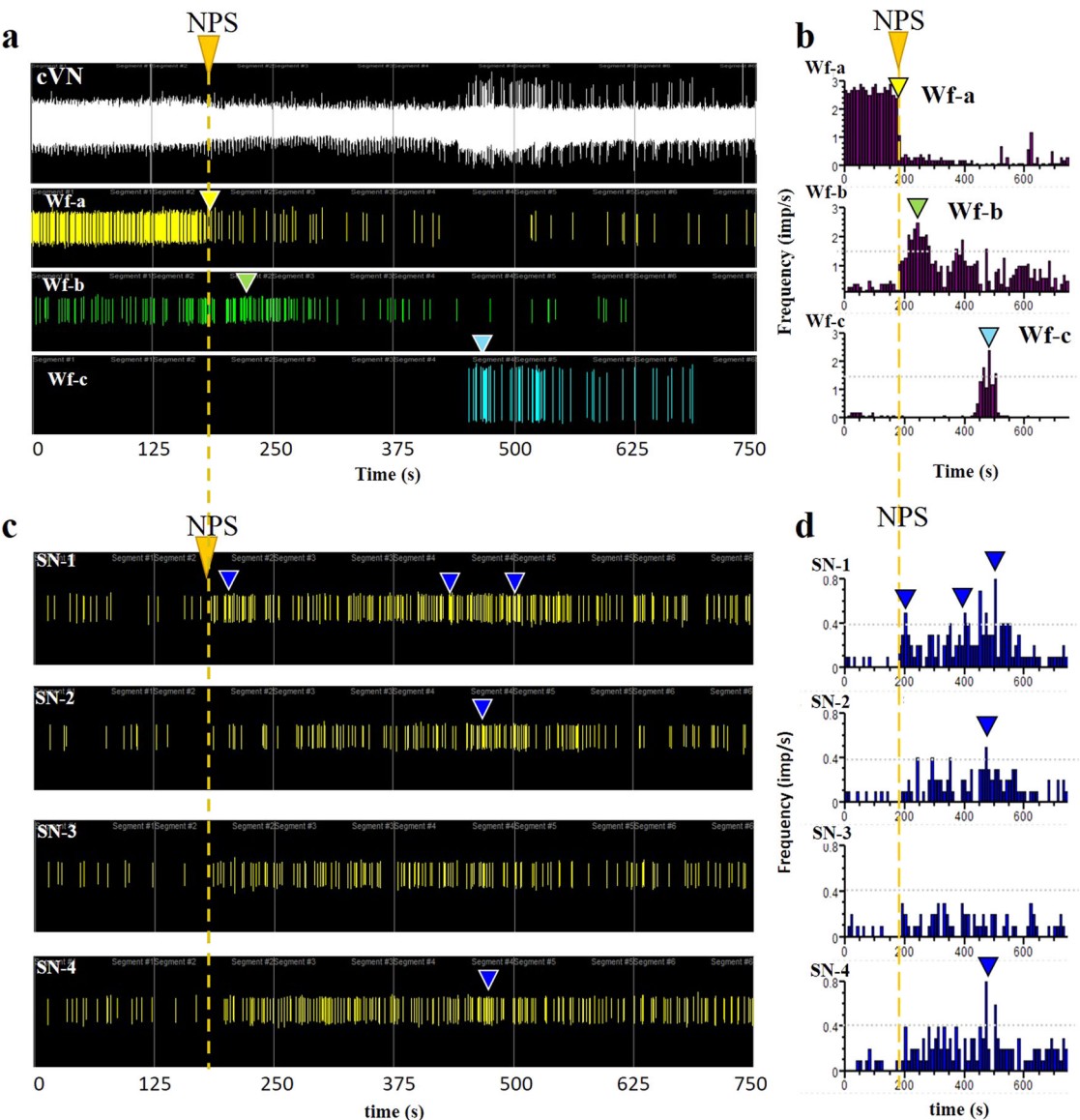

**Fig. 5 Hypotension inhibits cVN activity, followed by differential splenic branch activity. a** NPS administration (orange arrow and dashed line) causes a reduction in cVN activity (Wf-a, yellow arrow), and activation of two separate Wf (Wf-a and b, green and blue arrows, respectively), as demonstrated by the frequency histograms (**b**), changes in amplitude of the signal at 200 s compared to baseline was 1.6 ± 1.02 μV (Supplementary Fig. 4). **c** Simultaneous recording of neural activity in the four splenic branches showed a differential temporal response in frequency firing, with SN-1 following the loss of cVN Wfa, and SN-2 and SN-4 activated simultaneously with VN Wf-c. Delta in amplitude in response to the drop in mean arterial pressure (baseline vs. 200 s after NPS administration) where 4.6 ± 1.3, 3.4 ± 1.9, 0.9 ± 0.8, 2.8 ± 1.2, for SN-1–4, respectively. Tukey's multiple comparisons tests showed SN-1 vs. SN-3 **$P = 0.006$, alpha = 0.05, $n = 5$ (Supplementary Fig. 4). **d** Frequency histograms show specific time of increased activity over a 40% threshold (blue arrowheads in **c** and **d**). Representative waveforms were observed from $n = 6$ animals. NPS nitroprusside, cVN cervical vagus nerve, Wf waveform, SN splenic nerve.

(Fig. 8h, h´´). Negative controls (where SNVP was transected after viral administration, (Supplementary Fig. 8) did not present signal as expected (Fig. 8d´ and Supplementary Fig. 9f). Together, the data indicate that the brain has a direct anatomical and functional connection with the spleen through the VN, not previously recognized (Fig. 9).

## Discussion

Neural regulation of the immune system is critical for an effective and measured response to infection, trauma, or injury, and deregulated responses contribute to chronic inflammatory conditions[25]. Neuromodulation of the spleen inhibits the production of inflammatory cytokines and has substantial clinical

applications including rheumatoid arthritis, colitis, and sepsis[26–28]. However, the neural activity in the main splenic nerve trunk is only partially understood, and that of the terminal branches has remained unexplored due to technological challenges to place highly sensitive and flexible electrodes on the neurovascular plexus. Previous recordings from the main SN required the surgical isolation of the nerve from the blood vessels and resection of the nerve before entering the spleen, to facilitate placing the SN on traditional hook electrodes. Such studies reported a 27% increase in splenic vascular mechanoreceptor afferent activity over 5 min, in response to the increase in splenic venous pressure[16]. However, the simultaneous recording of nerve activity from all splenic branches has not been previously

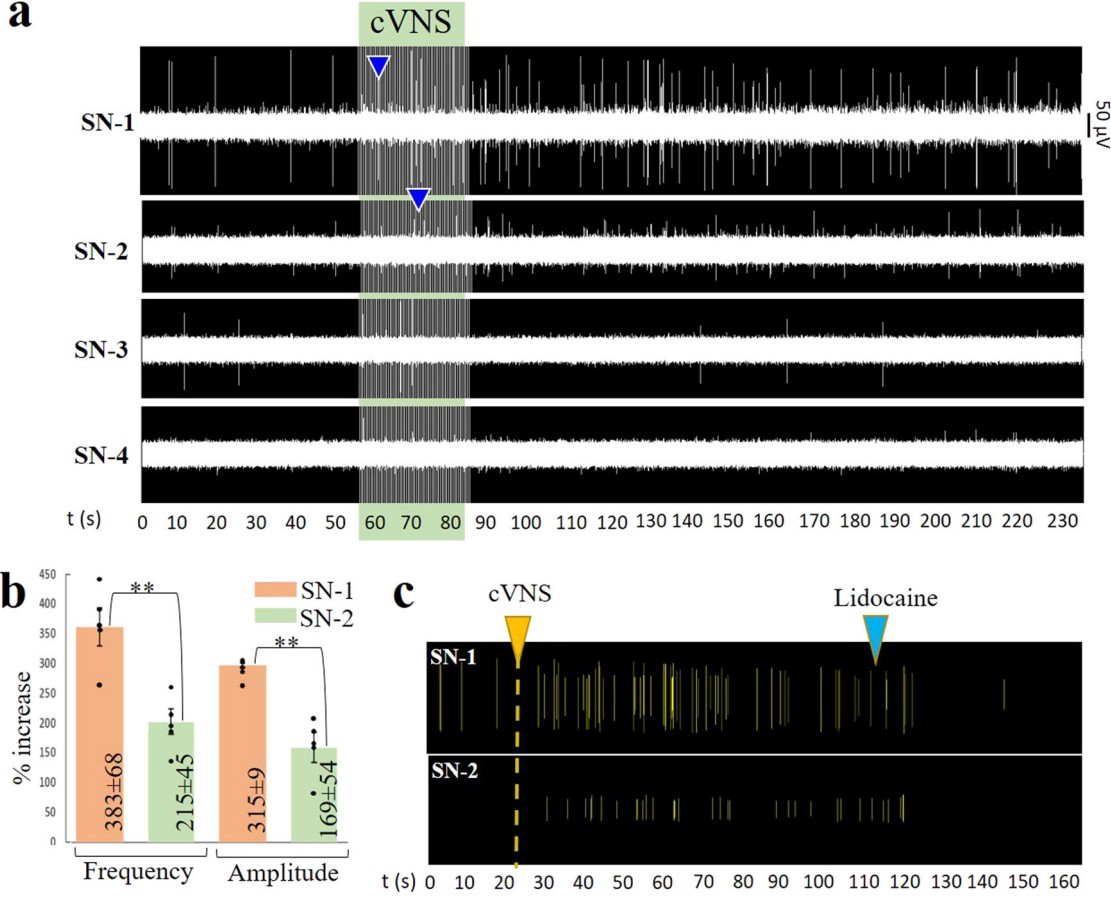

**Fig. 6 Differential neural activity in SN branches in response to cVNS. a** cVNS with 0.5 V, 30 s pulses evoked an immediate increase in SN-1 and SN-2 (blue arrowheads) and less in SN-3. No activity was evoked in SN-4. **b** Comparison between SN-1 and SN-2 spike frequency and amplitude signal increase, values were calculated at 60 s of stimulation and presented as mean ± SD for $n = 5$ animals; $**p < 0.01$. **c** Lidocaine (blue arrow) blocked the neuronal activity evoked by cVNS (yellow arrow), confirming its neuronal nature. SN splenic nerve, cVNS cervical vagus nerve stimulation, SD standard deviation.

achieved. In this work, we report the use of Pt-rGO fiber electrodes[22] as sutures (sutrodes), for the successful recording of neural activity from rat SNVP. Previously, this has been extremely difficult due to the small diameter of these fascicles in rodents (i.e., 50–100 μm nerves with 300–400 unmyelinated axons) and their location as they travel between blood vessels and are surrounded by fatty tissue. In the rat, terminal splenic fascicular branches are 8–12 times smaller and with ~1.75% of the number of axons compared to the cVN (600–800 μm; >20,000 axons). The combination of a small and fragile nerve, and the low axon content, makes the detection of CNAP from SNVP, a very challenging task. Here we report that the unique flexibility, mechanical strength, and electrochemical properties of the sutrode, allowed its placement over the SN terminal plexus, and the highly efficient recording of physiologically and electrically evoked CNAPs from the terminal SN fascicles.

Placing the sutrode on the cVN allowed the recording nerve activity evoked by hypotension and hypoxia. This capability is not surprising as recent multi-contact cuff electrodes[29] or intraneural carbon nanotube yarn electrodes were also used to record CNAP from the cVN[30]. However, the information recorded in those studies is limited, and often requires the use of advanced signal processing methods for the identification of clustered waveforms[31]. In this study, we were able to detect multiple waveforms associated with physiological changes directly. This is clearly shown in Fig. 5a where an immediate reduction in cVN activity was observed after NPS-induced vasodilation, suggesting the inhibition of afferent tonic activity, likely from low-threshold

vascular mechanoceptors[32]. Thus, the use of the sutrodes increased the quality and temporal resolution of the recorded neural information.

The unique flexibility and sensitivity of the sutrode allowed the efficient interfacing of the SNVP terminal branches and revealed that defasciculation from the main SN into the branches is not anatomically or functionally homogeneous. We learned that each terminal fascicle has unique neuron fiber content and responds selectively to specific physiological events. Since the vast majority of the nerve fibers in the splenic nerve are efferent sympathetic originating in the sympathetic intermediolateral column of the spinal cord and celiac-superior mesenteric ganglia[33,34], the evoked increase in CNAP activity by hypotension and hypoxia are likely sympathetic. However, the nature of the parasympathetic content and control has been a subject of debate and this study offers a new reconciliatory perspective.

It is well known that the spleen is under reflex modulation by the VN, mediated by the parasympathetic humoral arm, and afferent activity in response to ventral spleen compression[15]. Most recently, the SN was shown to be a necessary component of the neuroimmune reflex circuit with the VN; modulated by inflammatory cytokines such as TNFa and IL-1[6,35,36]. However, previous denervation and tract-tracing studies using traditional retrograde fluorescent molecules, have failed to confirm a direct innervation from the spleen to the VN[37,38], and electrophysiological studies have not demonstrated a direct VNS evoked activity in the SN[9,10]. The limitation of these previous studies is the use of silver wire electrodes for the SN recordings and the lack

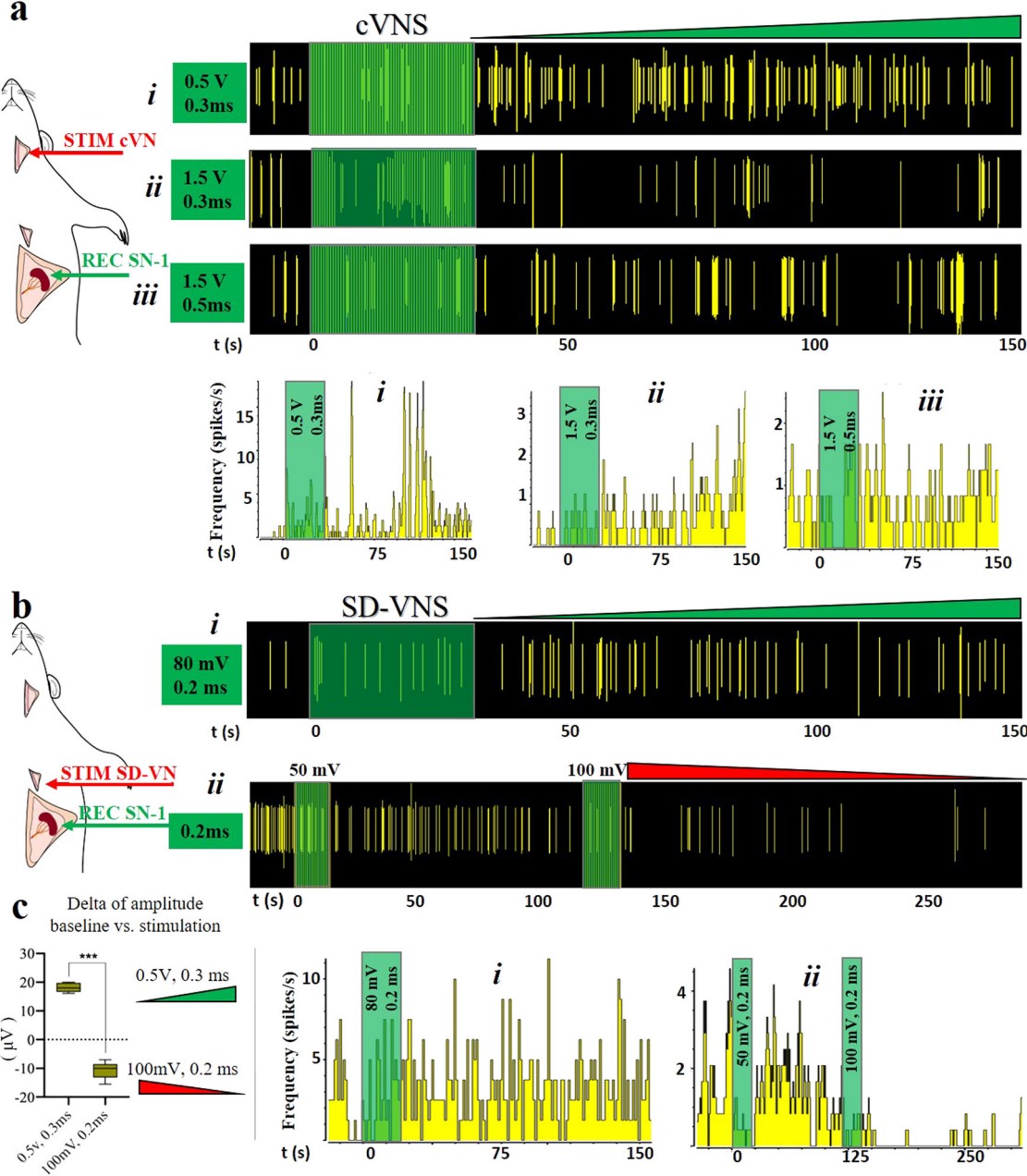

**Fig. 7 Inhibition of apical branch activity by cVNS. a** The activity in SN-1 after cVNS increased at (i) 0.5 V. (ii) At 1.5 V cVNS caused the opposite effect. (iii) Increasing the pulse to 0.5 ms, at 1.5 V had an intermediate effect. **b** SD-VNS with: (i) 0.2 ms stimulation at 80 mV caused an increase in activity, whereas stimulation of 50 and 100 mV (ii), inhibited SN-1 spontaneous activity. Green and red triangles indicate an increase and decrease in activity, respectively. Rate histograms are presented for each raster plot, representative waveforms are plotted, bin = 1 s. **c** Delta of changes on the amplitude of the signal (baseline vs. stimulation at 200 s) means ± standard deviation are: increase on 18.16 ± 1.6 and decrease on −10.68 ± 3.07 μV for 0.5 V, 0.3 ms and 100 mV, 0.2 ms, respectively. *T* test análisis, *p* < 0.0001 (*n* = 3 rats, 3–5 repetitions each). cVNS cervical vagus nerve stimulation, SD-VNS subdiaphragmatic vagus nerve stimulation, SN splenic nerve.

of resolution by the use of traditional (e.g. Fast-blue) tracers used in those studies[9]. In this report, the use of the highly sensitive and flexible Pt-rGO fiber electrodes allowed the recording of neural activity from intact splenic branches and showed evoked responses within 10 s after cVNS in some branches and delayed reflective activity in others (Fig. 6). The discovered heterogenous fiber content and diverse temporal response to physiological or electrically evoked activity in this study, suggest that from previous reports, electrophysiological recordings were limited by the positioning and sensitivity of traditional hook electrodes on the

SN, which only samples part of the neural activity. In addition, our viral tract tracing revealed clustered axonal fibers from the spleen in the cVN which are estimated to account for ~41% of the axons in SN-1, and <5% of SN-3 axons. The low number and variability of these VN-spleen axons may explain why these have been missed in previous reports using traditional axon tract-tracing methods in the main splenic nerve, with limited spatial resolution and electrophysiological recording with low temporal resolution[9]. Furthermore, SN-1 traced axons labeled perikarya in the DMN of the vagus, which agrees with studies using

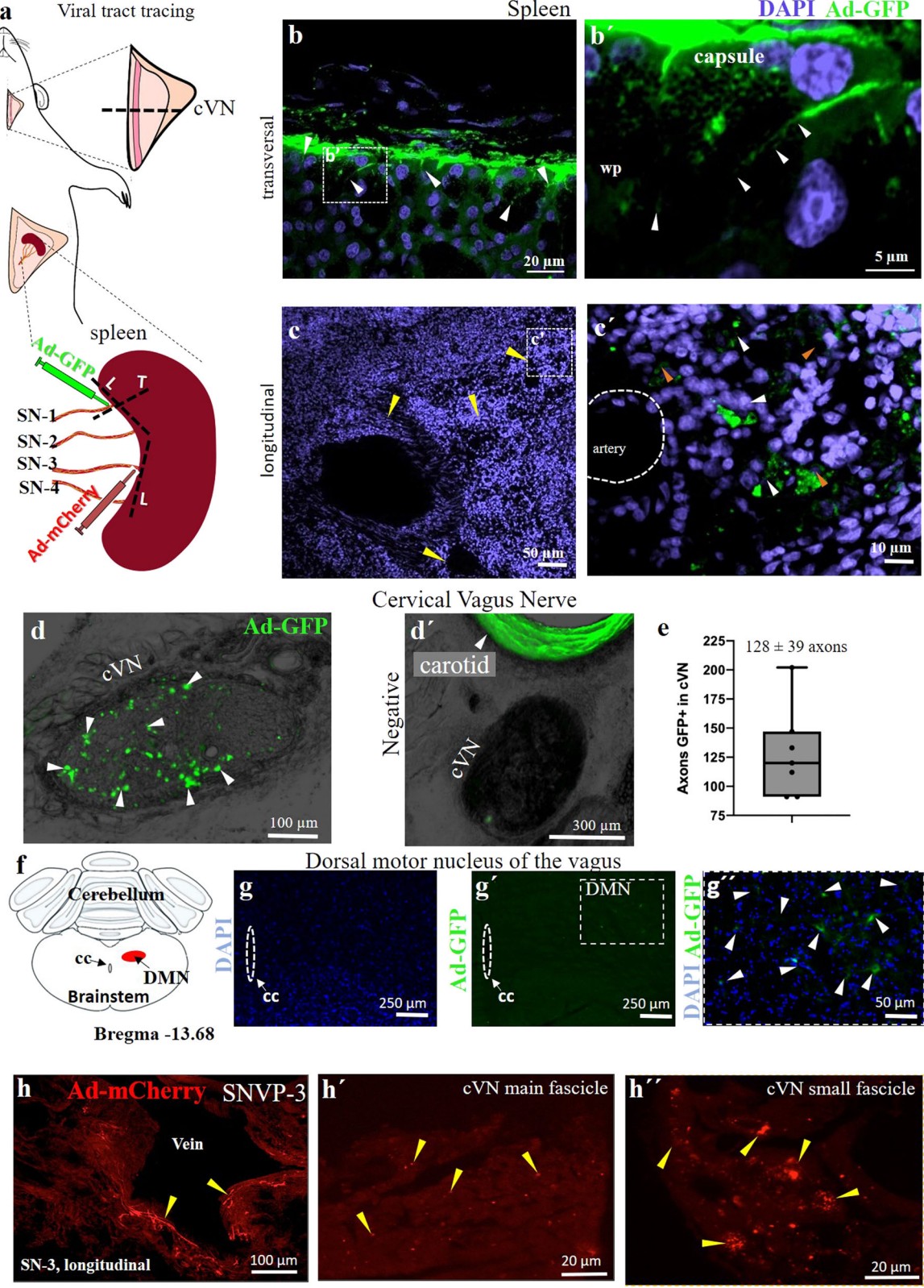

Choleratoxin-B and Pseudorabies virus directly injected into the spleen, where neurons in the DMN were found to directly innervate this organ[33,39].

The DMN in the medulla oblongata is known to be the site of origin of a pre-ganglionic parasympathetic efferent that innervates organs within the gastrointestinal tract, including the stomach and the pancreas[40]. Neurons in the DMN are innervated by neurons in the adjacent NTS, which receives most of the sensory afferent information from the VN and the nodose ganglia. The DMN and NTS are critical for the physiology of internal organs via the splenic-vagal reflex[40,41]. The combined anatomical and functional connection between the DMN and the spleen support a direct central parasympathetic control on the spleen through the splenic nerve where VN axons bypass the CG and fasciculate

**Fig. 8 Apical and basal SN branches have direct projections to the VN. a** Schematic of Ad-GFP injections in the SN-1, and Ad-mCherry in SN-3 for tract-tracing $n = 7$ animals positive and $n = 5$ negative controls. **b** Transverse histological sections of the spleen showing GFP+ cells (white arrows) in the splenic capsule and white pulp (wp). The white square is magnified as **b´**. **c** Transverse sections show blood vessels (yellow arrows) in proximity to GFP+ nerve fibers (white arrows in **c´**), often organized in clusters (orange arrows). **d** cVN highlighting GFP+ axons (white arrowheads, $n = 7$ animals). The negative control is presented in **d´**, where no fluorescent signal was observed; the carotid artery is auto-fluorescent by nature (white arrowhead) ($n = 5$). **e** Number of axons GFP+ quantified from $n = 7$ rats. **f** Schematic of coronal section of the brain used as a reference for **g–g´´**, where the dorsal motor nuclei of the vagus is pointed. One sample Wilcoxon test values show no significant difference in the distribution of the values from the mean at ∂¼ = 0.05. (in red). **g, g´** Cellular nuclei (blue) and GFP signal evidence cells in the DMN marked with AdGFP that was administrated in SN-1. Dotted line square magnification is presented in **g´´**, GFP+ cells are pointed with arrowheads. **h** mCherry+ axons (yellow arrowheads) adjacent to a vein in the spleen on SNVP-3; **h´** and **h´´** show mCherry+ axons in the cVN, main and a collateral fascicle. SN splenic nerve, cVN cervical vagus nerve, wp white pulp, DMN dorsal motor nucleus of the vagus, cc central canal, SNVP splenic neuro-vascular plexus.

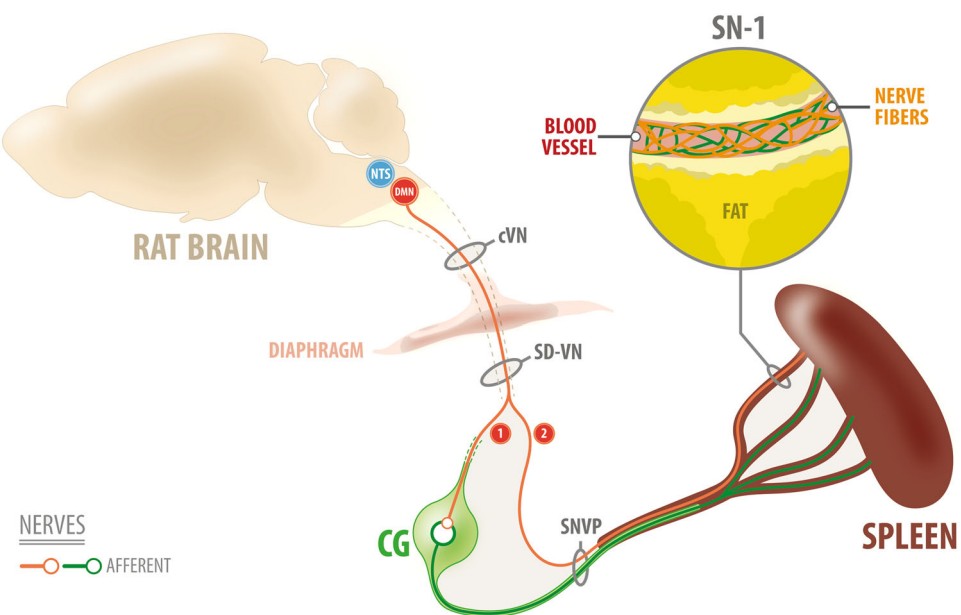

**Fig. 9 Brain-splenic control.** Schematic of proposed innervation of the spleen. (1) Neurons from the dorsal motor nucleus of the vagus (DMN) synapses in the celiac ganglia (CG) peripherally. The CG contains sympathetic norepinephrine fibers that innervate the spleen. (2) This work supports the presence of parasympathetic axons in the vagus nerve, that originate from neurons in the DMN and innervate the spleen directly. Projections from the DMN travel in the subdiaphragmatic vagus nerve (SD-VN), ending in terminal branches of the spleen (i.e., SN-1). SNVP splenic neurovascular plexus, cVN cervical vagus nerve.

into the SN-1 branch of the SNVP (Fig. 9). Further studies are warranted to extend these initial observations and address additional questions on the neurotransmitter and autonomic control of this major organ.

The potential clinical impact of selective neural stimulation of the splenic nerve terminal branches has been suggested recently, as stimulation of a terminal apical branch (not associated with a blood vessel; 650 µA, 100 µs pulses at 10 Hz), was shown to increase the release of Ach in the spleen and inhibit inflammatory cytokine secretion[14]. In this study, we expand and further define this possibility, by demonstrating that the type of terminal branch, and the specific electrical stimulation parameters used, can either increase or suppress the activity on the splenic fascicles, paving the way for a more selective and complete neuromodulation strategy for the spleen, and consequently, for the treatment of chronic inflammatory conditions.

Together, this study reports the use of a flexible, mechanically robust, and highly sensitive Pt-rGO sutrode for the interfacing of small nerve fascicles in the neurovascular plexus. Their use allowed the interrogation of small nerves, despite their intervascular location and small size, and evidenced a previously non-recognized anatomic–functional connection between the spleen and the VN. This approach will likely expand to the recording of other neurovascular plexus, particularly those in internal organs,

to inform, facilitate and enable their decoding and neuromodulation for bioelectronic applications.

## Methods

### Animal use

*Ethics statement.* All protocols and surgical procedures were designed to prevent animal discomfort and suffering at any moment. These were approved by The University of Texas at Dallas and the University of Houston, Institutional Animal Care and Use Committees (IACUC, protocols 14–09 and PROTO202000004, respectively), and follow the guidelines provided by the National Institute of Health (NIH).

**Surgical procedures.** A total of 46 Female Sprague Dawley rats (300–350 g; Charles River, Wilmington, MA) were used for the experiments. The animals were anesthetized with vaporized isoflurane (2%) in a constant oxygen flux (2 L min$^{-1}$) delivered by a calibrated vaporizer and maintained throughout the experiment. The animal temperature was maintained with an electrical warm pad. Anesthesia and vital signs were monitored constantly with standard methods throughout the experiment. Individual nerves studies included: -*Sciatic nerve (ScN)* exposed by a 4 cm longitudinal incision on the animal's hind limb from hip to knee following the femur. The biceps and quadriceps femoris were separated to visualize the ScN. -*The cervical vagus nerve (cVN)* was exposed by making a longitudinal medial incision in the anterior part of the neck at the cervical level, using the anterior sternum as reference. The sternomastoid muscle was separated in oblique orientation to the midline and the cVN was identified lateral to the carotid artery. -*Sub-diaphragmatic VN* (SD-VN) was isolated by a midline incision (2.5 cm) made on the abdominal wall, the stomach was moved slightly to expose the esophagus, where the SD-VN trunks were identified between the diaphragm and the gastric

cardia. -The spleen was identified in the upper left portion of the abdomen under the left part of the stomach. The splenic neurovascular terminal plexus was evidenced by gently lifting the spleen (Supplementary Video 3). For the ScN, cVN, and SD-VN, the connective tissue was carefully removed before electrode implantation. At the end of the experiments, an overdose of sodium pentobarbital (120 mg/kg) was used for euthanasia.

**Platinized graphene electrodes fabrication and validation.** The rGO fiber electrodes (aka sutrodes) were fabricated by the wet spinning of GO and characterized as previously reported[22]. GO was obtained by the thermal expansion (270–300 times) of intercalated graphite flakes grade (3772, Asbury Graphite Mills, USA) at $700 \pm 2\,°C$ under an inert atmosphere of Argon, using a vertical furnace equipped with a powder addition funnel. GO was reduced in hypo-phosphorous acid solution (50% in water, Sigma-Aldrich) at 80 °C for 24 h. After washing and drying, the rGO fibers (40 μm diameter) were sputter-coated with a 200 nm Pt layer. The Pt-rGO fibers were cut into 10 cm pieces by dipping into liquid nitrogen for about 1 min and then cut with a pre-cooled scissor. One end of the fiber was welded to a Pt wire, and 1 cm in the other end was covered with Parafilm, prior to coating with Parylene C using a deposition system (Specialty Coating System, PDS 2010 Labcoater). The Parafilm was then carefully removed to expose the Pt-rGO at the recording end of the fiber. The ultrastructure topology was assessed by scanning electron microscopy (SEM) using a microscope JEOL JSM-7500FA.

The electrical conductivity of Pt-rGO fibers was measured using a digital multimeter (Agilent 34401A) and the impedance was documented while making the sutrode into a knot (Supplementary Video 1). We then measured the baseline electrical noise recorded with the sutrode in saline solution and compared it to that of a commercial stainless-steel bipolar hook electrode (PBAA15100, FHC Inc., Bowdoin, ME) ($n = 6$).

**Electrode implantation.** To implant the sutrodes, the nerves were gently lifted using a blunted glass rod. A small piece of waxed parafilm was placed underneath the nerve to isolate it from surrounding tissue. Fine angled microsurgical forceps were used to wrap the Pt-rGO fiber electrode around the target nerve (SNVP-1 to 4, ScN or cVN) and an overhand knot was carefully tied, making sure that the circulation on the epineurium microvasculature was not obstructed. To stimulate the ScN, the cVN and the SD-VN, a hook electrode (PBAA15100, FHC Inc., Bowdoin, ME) was used. For the spleen surgery, saline humidified gauzes helped to separate and maintain hydrated surrounding visceral tissue.

**Mean arterial pressure and drug delivery.** The mean arterial pressure (MAP) was measured with a catheter in the femoral artery, and the drug delivery was administrated in the femoral vein. $n = 6$ rats were used for this study, 1–3 physiological tests per rat. A 1.0–1.5 cm incision in the medial aspect of the leg was made to expose both vessels, and a cannula (0.6 mm outer diameter) previously filled with heparinized saline (20 IU/mL) was inserted and secured to the muscle using 4.0 silk sutures. The arterial cannula was connected to a previously calibrated pressure transducer (AD-Instruments, MLT1199) coupled to a bridge amplifier and power supply modulus (AD-Instruments, FE221 and ML826, respectively) for continuously MAP evaluation. The venous cannula was coupled to an infusion system to administrate the vasoactive drug, nitroprusside (5 mg/mL, a bolus of 2.5 μg/g weight; 71778 Sigma Aldrich). Simultaneous nerve activity was recorded with the sutrodes placed on the cVN and SNVP-1 to 4. PowerLab data acquisition system (AD Instruments, Colorado Springs, CO) and LabChart Pro V8 (AD Instruments, Colorado Springs, CO) software were used to process and analyze the mean arterial pressure data.

**Electrophysiology and signal processing.** Graphene electrode fibers were coupled to a customized adapter that consists of clamps that connect to a 16 channels 20× gain, low noise miniaturized headstage (Plexon HST/16V-G20-LN) coupled to an Omniplex data acquisition system (SW-32, Plexon Inc., Dallas, TX). The electrophysiological activity was recorded at 40 kHz. A reference electrode consisted of a graphene fiber placed between the sternomastoid muscle and the skin or under the abdominal skin tissue for the VN or SN branches, respectively. Raw neural recordings were processed by a Butterworth, 4 pole high-pass frequency (250 Hz cutoff). Single waveforms were selected by placed a threshold above the noise level, manually, and confirmed by 2–3-dimensional PCA. Selected waveforms were processed with off-line sorter software (Plexon Inc, v3.3.5) and NeuroExplorer software (Nex Technologies, v 4.135) to determine the spike frequency distribution, interspike interval, and raster plots. A PlexStim electrical stimulator system (Plexon Inc.) was used to evoke neural activity in the ScN, cVN, and SDVN. Confirmation of the neural nature of the recorded activity was done by reduction of the signal after topical lidocaine application on the nerves, and elimination of the signal after euthanasia.

*ScN recordings.* The sciatic nerve was implanted with a sutrode and used to record evoked activity by (1) applying electrical stimulation with a bipolar hook electrode, placed 3 mm far in the proximal portion. 0.6 V symmetrical biphasic pulses, 0.3 ms width at 2 Hz with no interphase delay were applied. $n = 6$ rats were used for this study, 3–8 electrophysiological tests per rat. Or by (2) implanting 2 sutrodes in the

biceps muscle (~20 mm distanced from the recording site), using them as conductive sutures, and applying symmetrical biphasic pulses, 0.3 ms width at 2 Hz with no interphase delay, which elicited tissue contraction ($n = 6$ stimulation tests in one animal).

*VN stimulation.* Evoked activity in the SN-1 to 4 was evaluated by the stimulation of the cVN. $n = 3$ rats were used for this study, 3–5 tests per rat were performed with a total of 11 evoked responses. SN-1 to 4 were implanted with sutrodes and the cVN with a hook electrode for bipolar stimulation. 0.3 ms pulses of 0.5 V symmetrical biphasic pulses, 1 ms inter-pulse delay were applied by 30 s at 2 Hz. Furthermore, the study was focused on SN-1 activity. cVN was stimulated by 30 s in different experimental sets: (i) 0.3 ms pulses, 0.5 V; (ii) 0.3 ms pulses, 1.5 V, and (iii) 0.5 ms pulses, 1.5 V. Finally increased electrical trains of stimulation were applied to the cVN with a hook electrode, in this experiment a sutrode was included in the cVN, to simultaneously with the SN-1 to 4, record the evoked activity. 0.3 ms biphasic pulses, at 2 Hz, during 30 s were applied: 1.2, 1.4, 1.6, 1.8, and 2.0 V. 3.5 min were allowed between stimulation sets.

SD-VN stimulation effect on the SN-1 implanted with a sutrode was evaluated. $n = 4$ rats were used for this study, 3–4 experiments per rat were performed (total SD-VN stimulations = 13). Different stimulation parameters were applied with a hook electrode: 0.2 ms pulses, 2 Hz, at 50, 80 and 100 mV.

*Oxygen reduction.* Evoked neural activity in response to Oxygen restriction was evaluated by implanting sutrode recording electrodes on SNVP-1 to 4 or cVN. $n = 5$ rats were used for this study, 1–5 experiments per rat were performed (total physiological tests = 17). Baseline was recorded, followed by 2 min of Oxygen deprivation, elicited by closing the supply source. A hermetic face mask and extra latex seal in the borders between the mask and the head of the rat were used for this experiment in an effort to secure the Oxygen reduction. After opening the Oxygen flux, continuous recordings were collected. Our Oxygen deprivation system was validated by using a calibrated fiber-optic oxygen micro-sensor (OxyLite, Oxford-Optronix), which in saline bubbling Oxygen from the system used for the in-vivo experiments, allowed to determine that ~36 s were required for the Oxygen to drop after closing the Oxygen source.

**Ultrastructural analysis.** The ultrastructural analysis of the main splenic nerve and the neurovascular plexus of the spleen was assessed by transmission electronic microscopy (TEM). Four rats were transcardially perfused with physiological saline solution (NaCl, 0.9%), followed by a fixative solution that consists of 4% paraformaldehyde, 0.5% glutaraldehyde, in sodium arsenate (cacodylate) buffer (0.1 M, pH 7.4). The nerve sections of interest were attached to silk sutures for easy manipulation and dissected for post-fixation. The samples were post-fixed in 3.0% glutaraldehyde in cacodylate buffer (0.1 M pH 7.4), at 4 °C and processed for resin embedding and ultrathin sections. Standard toluidine blue staining was performed on 700 nm semi-thin sections to evaluate the structural organization. Ultra-thin sections of 60–70 nm were obtained using an ultramicrotome, mounted on copper grids, and coated with uranyl acetate and silver citrate for contrast. The sections were imaged using a transmission electron microscope (TEM, JEOL 1400 Plus, JEOL, USA). The images were obtained at high magnification and axon diameters were measured using Fiji Image J software (NIH).

**Adenoviral neuronal tract-tracing.** Two recombinant serotypes 5 adenovirus (Ad, dE1/E3) that codify for the expression of mCherry (Ad-mCherry) and GFP (Ad-GFP) proteins, under the control of the CMV promoter (Vector BioLabs, Philadelphia, USA) were used as axonal tracers (Supplementary Fig. 7). These were administered (~$3.0 \times 10^7$ viral particles/μl) with a Hamilton Neuro syringe (Hamilton, 1710) into the splenic terminal branches. Ad-GFP was used on SN-1 and Ad-mCherry on SN-3 ($n = 7$). Extreme efforts were made to prevent the spread of the tracers to the surrounding tissue and all the procedures were made using a surgical stereoscope. A parafilm was placed underneath the neurovascular-fat tissue during the administration, 15 min were allowed for the virus to be incorporated into the tissue after the administration, then a mini cotton tip was used to remove the viral solution and physiological interstitial liquid. Finally, a microneedle was used to make five washes with saline in order to maximize the removal of viral particles not incorporated in the tissue; each wash was followed by cleaning with a cotton tip, followed by the application of medical-grade silicone elastomer (Sylgard®) to prevent the spread of the tracer. As a negative control, the SNVP was transected and the extremes secured with surgical non-absorbable sutures, proximal and distal to the spleen after the viral administration (Supplementary Figs. 8 and 9, $n = 5$). The incision was closed, the muscle with 4.0 silk sutures, and the skin with metallic staples. Cephazolin (5 mg/kg, IM) and buprenorphine (1 mg/kg, SQ) were administrated subsequently as antibiotics and analgesic, respectively. After 6 days, the rats were intracardially perfused with physiological saline solution (NaCl, 0.9%), followed by fixative solution (4% paraformaldehyde in PBS). The VN attached to the carotid artery, and the SN branches were isolated and post-fixed in the same solution for 2 h. The samples were cryoprotected in increased sucrose gradients (10%, 20%, and 30%), embedded in OCT compound (Sekura Finetek, Torrance, CA, USA), and sectioned (45 μm thick) in a cryostat (Leica, CM3050s). The cellular nuclei were contrasted with 4′,6-

diamidino-2-phenylindole (DAPI, 1 mg/mL) and imaged in a confocal microscope (Nikon Eclipse Ti, Japan).

**Statistics and reproducibility**. We used the Kolmogorov–Smirnov normality test to determine the difference in axon diameter population in the different groups, followed by linear and non-linear regression of the normalized axon diameter distribution and used the slope $m$ to determine the statistical significance among the different splenic nerve terminal branches $p < 0.03$. We used a one-sample Wilcoxon test to evaluate the reproducibility of multiple measurements and distribution from the mean, $\alpha = 0.05$. Changes and comparisons on electrophysiological activity were determined using analysis of variance (ANOVA) or paired $t$-test. $p < 0.05$ was considered as significant. Statistical analysis was run on Prism 8 software (GraphPad, San Diego, CA).

**Reporting summary**. Further information on research design is available in the Nature Research Reporting Summary linked to this article.

## Data availability

All data generated and analyzed during this study are included in this article and in the Supplementary information. Datapoints used for graphics are provided as a supplementary file. The raw files are available from the corresponding author on reasonable request.

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

## Acknowledgements

The authors would like to thank the Australian National Fabrication Facility (ANFF) Materials node, for access to services and equipment, Prof. Muayyad Al-Ubaidi and Dr. Ryan Crane at the University of Houston for kindly allow with access to equipment for imaging. We acknowledge Prof. Rouhollah Ali Jalili for his contributions in the standardization and fabrication quality of the Pt-rGO fiber electrodes. Prof. Gordon Wallace and Kezhong Wang gratefully acknowledge funding from the Australian Research Council Centre of Excellence Scheme (Project Number CE 140100012). This study was supported by intramural funding at The University of Texas at Dallas and the University of Houston.

## Author contributions

Design of the research: M.R.-O., M.A.G.-G. In-vivo studies: M.R.-O., M.A.G.-G. TEM imaging: G.S.B. Fabrication and in-vitro characterization of Pt-rGO: G.G.W., K.W. Analyzed the results and reviewed the manuscript: M.R.-O., M.A.G.-G., G.S.B., K.W., G.G.W. Wrote the paper: M.R.-O., M.A.G.-G.

## Competing interests

The authors declare the following competing interests: M.R.-O. owns shares in RBI Medical, a medical device company. RBI Medical did not have any role in data collection, analysis, or the manuscript. The remaining authors have no competing interests.
