## [Peer Review File · Communications Biology]

Reviewers' comments:

Reviewer #1 (Remarks to the Author):

In this work, Gonzalez-Gonzalez and co-authors use graphene fiber electrodes to investigate the anatomy and physiology of spleen innervation. I appreciate the technical challenges of the experiments of stimulation/recording and in my opinion the data are convincing and deserve publication. I have two main comments to the work.

My first comment concerns the direct innervation of the vagus nerve to the spleen. If I understood this point correctly, the claim is that some axons of the vagus nerve cross the celiac ganglion and directly contact the spleen – and these should be the axons retrogradely labeled and shown in Figure 8. If this is the case, it should be possible to visualize the activation of VN nuclei in the brain stem upon (intense) stimulation of the spleen branches, i.e. by immunostaining for cFOS or other early genes and co-localizing this nuclear staining with retrogradely labeled cell bodies. I also suggest including a simple scheme either as main or as supplementary figure, showing this unconventional innervation path, which should help people who are not in the field to appreciate this point.

My second comment is about the downstream consequences of the selective stimulation of SN1-4. The authors show different activation patterns of the four branches in response to physiological challenges, i.e. hypotension and hypoxia. I wonder which the effect(s) could be of single branch activation on spleen physiology, i.e. whether localized ACh and/or pro-inflammatory cytokine release can be measured or visualized upon differential branch stimulation.

Reviewer #2 (Remarks to the Author):

In the manuscript entitled "Platinized graphene fiber electrodes revealed differential activity in terminal splenic neuromuscular plexi" MA Gonzalez-Gonzalez and colleagues describe a very fascinating way to better understand and elucidate the spleen physiology. The authors exploit miniaturised flexible graphene based electrode coated with Pt to record electrophysiological signals from the splenic nerve. The output highlights two important findings: i) the evidence for a direct vagus nerve control over the spleen and ii) the possibility to use such electrode to explore other neurovascular plexi.

The work is well written and organised, ranging from electrode fabrication and in vivo test, to ultrastructural analysis of the nerves. The electrophysiology data are well presented and the conclusions are fully supported by the data.

I have only minor comments to the authors:

1) In Materials and Methods, please add at page 31 line 356 in the GO used is commercial or not (in this case, please add a description of the synthesis method).

2) Page 36, line 456 (Adenoviral neuronal tracing), please add the scheme reporting the structure of the 2 adenoviruses used including the GFP and m-cherry sequence (the so called shuttle vector map)

3) In the discussion please avoid structure like "...for the first time..." , as at p.28 line 281 and p.30

line 321

Reviewer #3 (Remarks to the Author):

Following their recent publication in *Advanced Materials*, Gonzales-Gonzales et al. describe the use of graphene-fiber-(GF) based microelectrode with a thin platinum coating to record splenic nerve (SN) activity. Compared to previous recording strategies using cuff electrodes or intraneural carbon nanotube electrodes, GF-based microelectrode are highly sensitive and flexible, which make them a unique tool for recording peripheral nerve activity. [SEP]

Overall, this is a very interesting and exciting study that will move forward the field of peripheral nervous system physiology by providing a novel technical tool to record small peripheral nerves electrical activity. It also provides very important informations about the SN physiology, which has been underinvestigated.

However, the main drawback of the paper is the lack of analysis of interindividual variability. Data from different rats need to be pooled in order to perform statistics (in case the variability form one animal to another is too high, the authors might consider to normalize their data to baseline for each individual before plotting them). This will not only strengthen their conclusions about SN physiology but it will also support their claim about the superiority of their electrode compared to others.

More precisely, my comments are listed below. [SEP]

Major points:

- 1- For Figure 2: all 6 animals mentioned in the M&M need to be included in the graphs of panels b (MAP), d (count//bin) and e (breath/heart rates) with stats. Also the neurograms presented need to be quantified with stats demonstrated increased or decreased activity of the VN to support authors' conclusions.
- 2- Figure 3: How representative are panel 3d and e of all animals studied? Are the 4 rats mentioned in the M&M included in panel 3g and h? Please provide stats for Fig 3g.
- 3- Figure 4: How representative are the raster plots from one animal to another? Please provide pooled data of the 3 animals with stats.
- 4- Figure 5, 6 and 7 n = ? Once again, it is not known whether the results form one representative animal is presented or if it is the average of n = 3 as mentioned in the M&M.
- 5- The data provided in Figure 8 are quite provocative as the general consensus is that the VN and the SN are synaptically connected in the coeliac ganglion. Some authors have concerns about inadvertent spreading of tracers to abdominal vagal fibers that might explain artefactual staining of the VN (Anderson et al *Am J Physiol Heart Circ Physiol.* 2015 Dec 15;309(12):H2158. doi: 10.1152/ajpheart.00766.2015). While great cautious has been taken to avoid viral spreading as described in the M&M, one cannot rule out a possible leakage of the virus into the abdominal cavity. To overcome this technical difficulty, the authors might consider injecting the virus in an animal, where splenic nerve have been denervated as control. In that case the VN should not be labelled by reporter gene expression.

Minor points:

- 1- Fig 1h there might be a problem of y-axis scale. In that panel the amplitude of the CNAP is 100mV, which is huge.
- 2- There are 2 "Supplementary Fig. 2" and the supplementary figure showing the PCA of cVN spontaneous waveforms is lacking. Please check.

- 3- Page 9 line 124 "After 1 min of NPS administration, the mean arterial blood pressure (MAP) decreased by 50 mmHg approximately. This coincided temporally with (...) the appearance of CNAPs (...) at high frequency (Fig. 2b) likely from baroreceptor and cardiovascular afferents". This sentence might be confusing to readers since neither baroreceptor nor cardiovascular afferents are embedded in the VN. Baroreceptor and cardiovascular afferents might affect the VN activity indirectly. Please rephrase.
- 4- Fig. 2f y-axis need to be labelled.
- 5- Line 154 p16 "Fig. 3d-e" should be "Fig. 3d, e, g"
- 6- Fig. 3h is difficult to understand (black curves? Y-axis? Stats apply to which set of data?)
- 7- Line 172 page 18. "We used PCA on the unfiltered data to identify neural waveforms of spontaneous activity in a 6.5 ms window". These data should be provided as supplementary figure.
- 8- Line 178 p19 "Fig 1c" should be labelled "Fig 4c"
- 9- For figure 5a CNAP waveforms for the cVN and the SN should be provided as supplementary figure.
- 10- Figure 7 please provide quantification of the number of spikes to evidence the decrease or increase in SN activity.
- 11- For M&M: Very few details are given for neurogram recordings processing and analysis, which makes the replication of the study very difficult for other investigators.

Rebuttal Letter

We appreciate the careful and constructive review of our manuscript. In response to your comments, we added additional studies, updated relevant figures, and reviewed the text and references. The response to individual comments is provided point-by-point as follows:

Reviewer #1

In this work, Gonzalez-Gonzalez and co-authors use graphene fiber electrodes to investigate the anatomy and physiology of spleen innervation. I appreciate the technical challenges of the experiments of stimulation/recording and in my opinion the data are convincing and deserve publication. I have two main comments to the work.

My first comment concerns the direct innervation of the vagus nerve to the spleen. If I understood this point correctly, the claim is that some axons of the vagus nerve cross the celiac ganglion and directly contact the spleen – and these should be the axons retrogradely labeled and shown in Figure 8. If this is the case, it should be possible to visualize the activation of VN nuclei in the brain stem upon (intense) stimulation of the spleen branches, i.e. by immunostaining for cFOS or other early genes and co-localizing this nuclear staining with retrogradely labeled cell bodies. I also suggest including a simple scheme either as main or as supplementary figure, showing this unconventional innervation path, which should help people who are not in the field to appreciate this point.

R: We appreciate your constructive observations. We have included additional viral tract tracing studies and controls (denervating the spleen on terminal branching) that demonstrate the localization of spleen efferent innervation from neurons in the dorsal motor nucleus of the vagus (Fig. 8 and Supplementary Fig. 8-9, and additional references). Furthermore, we included an experiment that demonstrated the activation of splenic terminal branches when stimulating the sub-diaphragmatic vagus nerve in a range of 1-3 msec, which support the claim of direct communication (Supplementary Fig. 6). As suggested, we included a schematic with the innervation path described on this manuscript (Fig. 9).

My second comment is about the downstream consequences of the selective stimulation of SNI-4. The authors show different activation patterns of the four branches in response to physiological challenges, i.e. hypotension and hypoxia. I wonder which the effect(s) could be of single branch activation on spleen physiology, i.e. whether localized ACh and/or pro-inflammatory cytokine release can be measured or visualized upon differential branch stimulation.

R: We acknowledge this valuable suggestion on the functional and translational relevance. However, we are not currently equipped to answer this question directly. Instead, we have added this point in the discussion to indicate the need of additional studies that can uncover the nature of the neurotransmitters and modulators involved in the neural control of the spleen.

Reviewer #2

In the manuscript entitled "Platinized graphene fiber electrodes revealed differential activity in terminal splenic neuromuscular plexi" MA Gonzalez-Gonzalez and colleagues describe a very fascinating way to better understand and elucidate the spleen physiology. The authors exploit miniaturised flexible graphene based electrode coated with Pt to record electrophysiological signals from the splenic nerve. The output highlights two important findings: i) the evidence for a direct vagus nerve control over the spleen and ii) the

possibility to use such electrode to explore other neurovascular plexi. The work is well written and organized, ranging from electrode fabrication and in vivo test, to ultrastructural analysis of the nerves. The electrophysiology data are well presented and the conclusions are fully supported by the data.

I have only minor comments to the authors:

1) In Materials and Methods, please add at page 31 line 356 in the GO used is commercial or not (in this case, please add a description of the synthesis method).

R: As suggested, we have included details on source of GO and extra information on the synthesis of rGO.

2) Page 36, line 456 (Adenoviral neuronal tracing), please add the scheme reporting the structure of the 2 adenoviruses used including the GFP and m-cherry sequence (the so called shuttle vector map)

R: We have added the vector map with structure details as Supplementary Fig 7.

3) In the discussion please avoid structure like "...for the first time..." , as at p.28 line 281 and p.30 line 321

R: Thank you for your suggestion, we have edited the text and avoided such descriptions.

Reviewer #3

Overall, this is a very interesting and exciting study that will move forward the field of peripheral nervous system physiology by providing a novel technical tool to record small peripheral nerves electrical activity. It also provides very important informations about the SN physiology, which has been underinvestigated. However, the main drawback of the paper is the lack of analysis of interindividual variability. Data from different rats need to be pooled in order to perform statistics (in case the variability from one animal to another is too high, the authors might consider to normalize their data to baseline for each individual before plotting them). This will not only strengthen their conclusions about SN physiology but it will also support their claim about the superiority of their electrode compared to others.

More precisely, my comments are listed below.

Major points:

1- For Figure 2: all 6 animals mentioned in the M&M need to be included in the graphs of panels b (MAP), d (count/bin) and e (breath/heart rates) with stats. Also the neurograms presented need to be quantified with stats demonstrated increased or decreased activity of the VN to support authors' conclusions.

R: Bar graphs were added to Fig. 2 to represent the data from the animals included in each study. This information is now clearly indicated in the text and figure legend.

2- Figure 3: How representative are panel 3d and e of all animals studied? Are the 4 rats mentioned in the M&M included in panel 3g and h? Please provide stats for Fig 3g.

R: The data is representative and include the 4 animals as mentioned in M&M. Statistical analysis for Fig. 3g is now included.

3- Figure 4: How representative are the raster plots from one animal to another? Please provide pooled data of the 3 animals with stats.

R: The plots are representative and we added a graph that shows the data from the 3 animals to Fig. 4.

4- Figure 5, 6 and 7 n = ? Once again, it is not known whether the results from one representative animal is presented or if it is the average of n = 3 as mentioned in the M&M.

Report statistics (1-4)

R: We appreciate the comment by the reviewer. The number of animals are: 6 in Fig. 5, 5 in Fig. 6, and 3 for Fig. 7. Figures were modified to add population data, and the number of animals added to the figure legends.

5- The data provided in Figure 8 are quite provocative as the general consensus is that the VN and the SN are synaptically connected in the coeliac ganglion. Some authors have concerns about inadvertent spreading of tracers to abdominal vagal fibers that might explain artefactual staining of the VN (Anderson et al *Am J Physiol Heart Circ Physiol.* 2015 Dec 15;309(12):H2158. doi: 10.1152/ajpheart.00766.2015). While great cautious has been taken to avoid viral spreading as described in the M&M, one cannot rule out a possible leakage of the virus into the abdominal cavity. To overcome this technical difficulty, the authors might consider injecting the virus in an animal, where splenic nerve have been denervated as control. In that case the VN should not be labelled by reporter gene expression.

R: We appreciate your important suggestion to increase the confidence in our data. As requested, we included two additional rats for tract tracing. One more was added as a negative control where the splenic terminal branch was transected prior to tracing (panel g in Supplementary Fig. 9) to eliminate the possibility of false positive labeling due to leakage of viral particles. The additional results confirmed our previous findings (Fig 8 and Supplementary Fig 8-9). We also confirmed GFP+ neurons in the dorsal motor nucleus of the vagus, which were not labeled in the negative control. These experiments reinforced our previous conclusions. The discussion was modified to put this new information in context and additional references included.

Minor points (Review and make corrections):

1- Fig 1h there might be a problem of y-axis scale. In that panel the amplitude of the CNAP is 100mV, which is huge.

R: Thank you for your observation, that was a regretful mistake and Fig. 1h is now corrected.

2- There are 2 “Supplementary Fig. 2” and the supplementary figure showing the PCA of cVN spontaneous waveforms is lacking. Please check.

R: We corrected text and reviewed Supplementary Fig. 2 to show the PCA of two spontaneous waveforms.

3- Page 9 line 124 “After 1 min of NPS administration, the mean arterial blood pressure (MAP) decreased by 50 mmHg approximately. This coincided temporally with (...) the appearance of CNAPs (...) at high frequency (Fig. 2b) likely from baroreceptor and cardiovascular afferents”. This sentence might be confusing to readers since neither baroreceptor nor cardiovascular afferents are embedded in the VN. Baroreceptor and cardiovascular afferents might affect the VN activity indirectly. Please rephrase.

R: We apologize for the confusing sentence. It is widely accepted that parasympathetic vagal fibers densely innervate the atria, SA and AV nodes. However, the reviewer is correct in noting that given the time of the response, the effect in vagal activity is likely indirect. This is now clarified in the text and two references added.

4- Fig. 2f y-axis need to be labelled.

R: Thank you for your observation, we labeled the y-axis in that Figure.

5- Line 154 p16 “Fig. 3d-e” should be “Fig. 3d, e, g”

R: We fixed this oversight.

6- Fig. 3h is difficult to understand (black curves? Y-axis? Stats apply to which set of data?)

R: The graph was modified to improve clarity and the text expanded to explain the detail of the statistical analysis.

7-8- Line 178 p19 “Fig 1c” should be labelled “Fig 4c”

R: Thank you for your observation, we fixed this.

9- For figure 5a CNAP waveforms for the cVN and the SN should be provided as supplementary figure.

R: Thank you for your suggestion, we have included CNAP waveforms for cVN and SN as Supplementary Fig. 4

10- Figure 7 please provide quantification of the number of spikes to evidence the decrease or increase in SN activity.

R: We acknowledge your suggestion, we included rate histograms for each panel to represent the number of spikes/sec through the experiments.

11- For M&M: Very few details are given for neurogram recordings processing and analysis, which makes the replication of the study very difficult for other investigators.

R: We regret this oversight. Information on signal processing of neural recording data is now included.

We are thankful for the careful and detailed review on the manuscript, and hope that you find the revised manuscript suitable for publication in Communications Biology.

Best Regards,

Mario Romero-Ortega, Ph.D. Cullen Professor
Departments of Biomedical Engineering and Biomedical Sciences
Colleges of Engineering and Medicine
Health 2 Building, Rm: 8005-8007.
University of Houston

Reviewers' comments:

Reviewer #1 (Remarks to the Author):

I appreciate the improvements made to the work, which satisfactorily address the points I previously raised. I believe the manuscript deserves publication.

Reviewer #2 (Remarks to the Author):

The authors fully addressed all the concerns. In my opinion, the manuscript is now ready for publication as it is. Congrats, very nice work.

Reviewer #3 (Remarks to the Author):

I appreciate the efforts that were made to answer my points and acknowledge a significant improvement of the manuscript compared to the initial version. However, some of my recommendations have still not been addressed (see below)s.

4- Figure 5, 6 and 7 n = ? Once again, it is not known whether the results from one representative animal is presented or if it is the average of n = 3 as mentioned in the M&M.

Report statistics (1-4)

R: We appreciate the comment by the reviewer. The number of animals are: 6 in Fig. 5, 5 in Fig. 6, and 3 for Fig. 7. Figures were modified to add population data, and the number of animals added to the figure legends.

-> Figure 5 has not been changed as requested. Only one representative animal is presented. No statistics are provided.

Figure 6 is ok but "n = 5 samples" need to be detailed. What does "samples" means in this case: number of animals? Same animal but different recordings?

For figure 7, once again results from different animals are not presented and no stats are provided to substantiate the claims presented in the text.

5- The data provided in Figure 8 are quite provocative as the general consensus is that the VN and the SN are synaptically connected in the coeliac ganglion. Some authors have concerns about inadvertent spreading of tracers to abdominal vagal fibers that might explain artefactual staining of the VN (Anderson et al *Am J Physiol Heart Circ Physiol.* 2015 Dec 15;309(12):H2158. doi: 10.1152/ajpheart.00766.2015). While great cautious has been taken to avoid viral spreading as described in the M&M, one cannot rule out a possible leakage of the virus into the abdominal cavity. To overcome this technical difficulty, the authors might consider injecting the virus in an animal, where splenic nerve have been denervated as control. In that case the VN should not be labelled by reporter gene expression.

R: We appreciate your important suggestion to increase the confidence in our data. As requested, we included two additional rats for tract tracing. One more was added as a negative control where the splenic terminal branch was transected prior to tracing (panel g in Supplementary Fig. 9) to eliminate the possibility of false positive labeling due to leakage of viral particles. The additional results confirmed our previous findings (Fig 8 and Supplementary Fig 8-9). We also confirmed GFP+ neurons in the dorsal motor nucleus of the vagus, which were not labeled in the negative control.

These experiments reinforced our previous conclusions. The discussion was modified to put this new information in context and additional references included.

-> Once again very provocative conclusion of a direct VN connection to the spleen that needs to be backed up by unequivocal data. Since some staining is observed in the negative control presented, more animals need to be included and results have to be quantified with a number of animal sufficient to reach statistical significance.

Also a representative image of staining for GFP in DMN is missing for negative controls in supp figure 9.

-> Minor point:

Figure 4: "... for all SN, n=3,)" please correct

July 26, 2021

Chao Zhou, PhD
Editorial Board Member
Communications Biology

Dear Dr. Zhou,

We acknowledge the comprehensive input provided from the reviewers, that improve the content and quality in our manuscript. As response to your suggestions, we performed new experiments for viral tract tracing, including negative controls by transecting neuronal targets, and added more quantitative analysis as suggested by Reviewer 3. Individual observations were addressed as summarize below:

Reviewer #1:

I appreciate the improvements made to the work, which satisfactorily address the points I previously raised. I believe the manuscript deserves publication.

Reviewer #2:

The authors fully addressed all the concerns. In my opinion, the manuscript is now ready for publication as it is. Congrats, very nice work.

Reviewers 1&2: We appreciate your detailed review that improved the quality of our manuscript.

Reviewer #3:

1.- Figure 5, 6 and 7 n = ? It is not known whether the results form one representative animal is presented or if it is the average of n = 3. Report statistics (1-4).

R: We appreciate the comment by the reviewer. The number of animals is: 6 in Fig. 5, 5 in Fig. 6, and 3 for Fig. 7. Figures were modified to add population data, and the number of animals added to the figure legends and is detailed in the manuscript. Statistical significance is now provided for Figures 1-4.

2.- Figure 5 has not been changed as requested. Only one representative animal is presented. No statistics are provided.

R: Fig. 5 is from a representative animal, from a total of 6 animals evaluated. Of these animals we observed a reduction of signal in the VN in 5 animals. The average reduction in signal level has been included in the text. The figure is already busy and was not modified, however we prepared a graphic and a descriptive table, included as Supplementary Figure 4.

3.- Figure 6 is ok but “n = 5 samples” need to be detailed. What does “samples” means in this case: number of animals? Same animal but different recordings?

R: Yes. These were 5 different animals. That is now clarified in the figure legend.

4.- For figure 7, once again results from different animals are not presented and no stats are provided to substantiate the claims presented in the text.

R: We improved the Figure by adding in panel c), a graphic that compares changes in the activity with the parameters that increase and decrease the activity in the splenic nerve 1. Statistics are indicated in the Figure, legend and in the manuscript. We clarified the numbers in the Figure legend as: n=3 rats, 3-5 replicates per animal.

5- Once again very provocative conclusion of a direct VN connection to the spleen that needs to be backed up by unequivocal data. Since some staining is observed in the negative control presented, more animals need to be included and results have to be quantified with a number of animal sufficient to reach statistical significance. Also a representative image of staining for GFP in DMN is missing for negative controls in supp figure 9.

R: We appreciate your important suggestion to increase the confidence in our data. As requested, we increased our number of animals to 7 rats for positive labeling, and 5 for negative controls. Our negatives where transected in the splenic branch prior to tracing, we added an extra step, covering the target area with biomedical grade silicon, and leaving parafilm underneath to completely eliminate the risk of virus leaking. A new Supplementary Fig. 8 is now added to cover the method and results. The additional experiments confirmed our previous findings (Fig 8 and Supplementary Fig 9). We also quantified the number of axons GFP+ observed in the vagus nerve (Figure 8, panel e). We confirmed GFP+ neurons in the dorsal motor nucleus of the vagus, which were not labeled at all, in the negative control (presented in Supplementary Fig 9, panel i). Together, we can state with confidence that there are approximately 128 axons out of 350 counted by EM in the SN-1 branch, that directly connect ventral motor neurons in the brainstem, with the spleen.

6.- Minor point: Figure 4: "... for all SN, n=3,)" please correct

R: thank you, we corrected the typo

Additional comments: We since became aware that "plexi" was incorrectly used and that "plexus" is the proper term. Thus, we have now use plexus in title and through the manuscript.

We are very thankful for the careful and detailed review on the manuscript, and hope that you find the revised manuscript suitable for publication in Communications Biology.

Best Regards,

Mario Romero-Ortega, PhD.